# Solid-Phase Synthesis of Red Fluorescent Carbon Dots for the Dual-Mode Detection of Hexavalent Chromium and Cell Imaging

**DOI:** 10.3390/bios12060432

**Published:** 2022-06-20

**Authors:** Jinshuang Hu, Xin Wang, Hua Wei, Lei Zhao, Boxuan Yao, Caiyun Zhang, Jiarui Zhou, Jian Liu, Shenghong Yang

**Affiliations:** 1Shandong Provincial Key Laboratory of Molecular Engineering, School of Chemistry and Chemical Engineering, Qilu University of Technology (Shandong Academy of Sciences), Jinan 250353, China; hjs05265969@163.com (J.H.); wx19971221@163.com (X.W.); wh17861406068@163.com (H.W.); ybx20021218@163.com (B.Y.); zhangcaiyun209@163.com (C.Z.); zjr13361523713@163.com (J.Z.); 2Key Laboratory of Biotechnology and Bioengineering of State Ethnic Affairs Commission, Biomedical Research Center, Northwest Minzu University, Lanzhou 730030, China; 3Institute of Advanced Materials, Jiangxi Normal University, Nanchang 330022, China; jianliu18@jxnu.edu.cn

**Keywords:** solid-phase synthesis, R-CDs, scanometric and fluorescent, Cr(VI) detection

## Abstract

The excellent optical properties and biocompatibility of red fluorescence carbon dots (R-CDs) provide a new approach for the effective analysis of hexavalent chromium Cr(VI) in environmental and biological samples. However, the application of R-CDs is still limited by low yield and unfriendly synthesis route. In this study, we developed a new type of R-CDs based on a simple and green solid-phase preparation strategy. The synthesized R-CDs can emit bright red fluorescence with an emission wavelength of 625 nm and also have an obvious visible light absorption capacity. Furthermore, the absorption and fluorescence signals of the R-CDs aqueous solution are sensitive to Cr(VI), which is reflected in color change and fluorescence quenching. Based on that, a scanometric and fluorescent dual-mode analysis system for the rapid and accurate detection of Cr(VI) was established well within the limit of detection at 80 nM and 9.1 nM, respectively. The proposed methods also possess high specificity and were applied for the detection of Cr(VI) in real water samples. More importantly, the synthesized R-CDs with good biocompatibility were further successfully applied for visualizing intracellular Cr(VI) in Hela cells.

## 1. Introduction

Heavy metals are a kind of nondegradable, persistent pollutants, and the related research has been the focus and difficulty of biological and environmental science [1,2]. As a typical dangerous metal, hexavalent chromium (Cr(VI)) is widely used in electroplating, leather tanning, textile, chemical fertilizer, stainless steel, welding, and wood preservative industries [3]. Untreated discharge of industrial wastewater and waste residue containing chromium into the environment can lead to the enrichment of Cr(VI) through the food chain, which not only causes serious environmental pollution, but also seriously threatens human life safety [4]. More importantly, a trace amount of Cr(VI) is sufficient to cause bronchitis, skin ulcers, liver, kidney, and nerve tissue damage, and even cancer, [5,6]. The International Agency for Research on Cancer (IARC) classifies chromate as a class substance (human carcinogen). The World Health Organization (WHO) has determined that the maximum pollution level of Cr(VI) in drinking water is 50 µg/L [7], and the United States Environmental Protection Agency (USEPA) has stipulated that the permissible chromium content in drinking water cannot exceed 100 µg/L [8]. In general, the concentration of Cr(VI) in drinking water has been regulated more strictly in recent years. Therefore, real-time and rapid detection of Cr(VI) in environmental water samples is of great significance for environmental protection and human health and safety.

At present, the common detection methods of Cr(VI) mainly include atomic absorption spectrometry [9], inductively coupled plasma atomic emission spectrometry [10], inductively coupled plasma mass spectrometry [11], and the electrochemical method [12,13], etc. These methods are all effective in the accurate determination of Cr(VI), but they still cannot meet the requirements of simple and real-time detection of Cr(VI) due to the limitation of large or expensive equipment, complex operations, cumbersome pretreatment, and time consumption [14]. Encouragingly, the scanometric/fluorescent dual-readout method is being advocated due its simplicity and multifaceted applicability. A scanometric analysis system can realize low-cost rapid quantification of the target analyte with no need for an optical instrument [15,16], and the fluorescent method provides a powerful guarantee for the trace amount analysis and in vivo imaging analysis [17]. Therefore, the development of new scanometric/fluorescent dual-mode analysis systems aimed at Cr(VI) determination is urgent for solving the shortcomings of traditional means.

Carbon dots (CDs), as one of the most dynamic research objects in optical materials, not only inherit the excellent optical properties of traditional semiconductor quantum dots, but also make up for the shortcomings of traditional fluorescent materials in cytotoxicity, environmental, and biological risks [18]. CDs exhibit a broad prospect in environmental and biological analysis applications due their inherent attributes of good water solubility, stable chemical properties, and surface tunability [19,20,21]. Specially, red fluorescence CDs (R-CDs) possess unique advantages for bioimaging and biosensing application, which can effectively eliminate the interference of background fluorescence from biological tissue [22]. However, current R-CDs still suffer from low photoluminescence efficiency and product yield, and the synthesis of them also usually cannot remove their dependence on organic solvent and inorganic acid [23,24,25,26]. The solid-phase synthesis method was advocated for in the preparation of CDs to overcome the deficiencies of traditional solvothermal and microwave methods because of its simplicity, solvent-free, easy operation and repetition. Unfortunately, the most reported CDs obtained by solid-phase synthesis can only emit blue or green fluorescence, which compromises on the indepth application of CDs in the field of bioscience [27,28,29]. Therefore, it is still a challenge to develop R-CDs based on the simple solid-phase synthesis method and further promote the practical application of CDs.

In this work, we reported a simple solid-phase synthesis strategy for the preparation of R-CDs using o-phenylenediamine and aniline hydrochloride as raw materials. The reaction route only needs 2 h without solvent involved, which is consistent with green chemistry and further promotes the practicability of CDs. The obtained R-CDs can emit bright red fluorescence with a quantum yield of 22.9%, and the product yield can reach 42.4%. For another, the R-CDs have a good visible light absorption property and the aqueous solution appears blue. Furthermore, the developed R-CDs possess good stability and abundant surface-active functional groups. Cr(VI) can induce R-CDs to create obvious changes in both solution color and fluorescence intensity, and a high-sensitivity and high-specificity scanometric/fluorescence dual-mode analysis system for Cr(VI) was established. Actual environmental water samples analysis was performed to verify the simplicity and reliability of the scanometric mode, while the fluorescence mode was applied for the visual imaging of intracellular Cr(VI) (Figure 1). This study provides a new idea for the green and rapid synthesis of CDs and the development of new analytical methods.

## 2. Materials and Methods

### 2.1. Materials

Aniline hydrochloride, o-phenylenediamine (OPD) and sodium chromate (Na_2_CrO_4_) were purchased from Aladdin Biological Technology Co., Ltd. (Shanghai, China). Glycine was obtained from McLean Biological Technology Co., Ltd. (Shanghai, China). Hydrochloric acid (HCl) was purchased from Yantai Yuandong Fine Chemicals Co., Ltd. (Yantai, China). All other reagents were of analytical reagent grade and used without any further purification. Ultrapure water with a resistivity of 18.25 MΩ·cm (UPR-II-40L, Sichuan, China) was used in the whole experiment.

### 2.2. Characterization

The ultraviolet-visible absorption and photoluminescence (PL) spectra of R-CDs were measured by an UV2800S UV-visible spectrophotometer (Shanghai Hengping Scientific Instrument Co., Ltd., Shanghai, China) and an F97Pro fluorescent spectrophotometer (Shanghai Lengguang Technology Co., Ltd., Shanghai, China), respectively. The size distribution and morphology of the prepared R-CDs were characterized by a Tecnai G2 F30 transmission electron microscopy (TEM) with a 200 kV accelerating voltage (FEI, Hillsboro, OR, USA) and a BI-200SM dynamic light scattering (DLS) particle size analyzer (Brookhaven, NY, USA). X-ray powder diffraction (XRD) spectrum of R-CDs was recorded using a D8-ADVANCE diffractometer (Bruker, Saarbrucken, Germany). The molecular structure of R-CDs was characterized by a NEXUS 670 Fourier transform infrared spectroscopy (FTIR, Nicolet, WI, USA) and an EscaLab Xi+ X-ray photoelectron spectroscopy (XPS, Thermo Fisher Scientific, MA, USA). The MTT test was performed on an RT-6100 enzyme-mark analyzer (Shenzhen, China). The fluorescent imaging photographs of the cells were taken using Axioscope A 1 POL fluorescence microscope (ZEISS, Oberkochen, Germany).

### 2.3. Preparation of R-CDs

The R-CDs were synthesized using OPD and aniline hydrochloride as raw materials through a one-step solid-phase synthesis method. Typically, 0.15 g OPD and 0.3 g aniline hydrochloride were grinded thoroughly in an agate mortar, and the resulting mixture was transferred to a 50 mL Teflon-lined stainless steel autoclave and heated in 200 °C for 2 h. After cooling to room temperature, the product was dispersed in ethanol and dialyzed for 48 h (molecular weight cut-off 7000). The solution was further dried under vacuum conditions, and the obtained solid powder was R-CDs.

### 2.4. Determination of Cr(VI)

The colorimetric and fluorescence detection of Cr(VI) was performed in a 5 mM glycine-hydrochloric acid buffer solution with pH at 2.0 and 3.0, respectively. Amounts of 30 μL Cr(VI) with different concentrations were added to the R-CDs solutions (400 μg/mL), respectively, and incubated for 10 min. The scanometric analysis was carried out in a 96-well plate, and RGB values of all solution photos were analyzed by Image J. For fluorescence analysis, the PL spectra were recorded at 560 nm excitation and the PL intensities were recorded at 625 nm. The selectivity of the proposed methods to Cr(VI) was measured by adding various anions and cations, and the detection conditions were the same as described above. The practicability of the methods for Cr(VI) analysis was verified through real water sample analysis and cell imaging.

### 2.5. Cellular Imaging

Hela cells were inoculated on 6-well plates that contained 100 U·mL^−1^ of penicillin and streptomycin and 10% fetal bovine serum, then cultured in a 5% CO_2_ humidified incubator at 37 °C for 48 h. Then, 400 μg·mL^−1^ R-CDs in glycine-hydrochloric acid buffer solution (5 mM, pH = 3) was used to incubate Hela cells for 20 min in the same culture conditions, and the cells were imaged on a fluorescent microscope after washing twice with fresh buffer solution. Then, different concentrations of Cr(VI) (0, 5, 15, 30 μM) were added and incubated for 10 min. Before imaging, they were washed twice with fresh buffer solution to remove excessive Cr(VI). The images were captured again using a fluorescence microscope under green laser light excitation.

## 3. Results and Discussion

### 3.1. Characterization of R-CDs

The R-CDs were obtained through the facile solid-phase synthesis method, which can avoid the involvement of toxic solvents effectively and further promote the commercial application of R-CDs. The synthesized R-CDs were characterized in detail. As shown in Figure 1a, the UV-visible absorption spectrum of R-CDs displayed three obvious absorption peaks at 285 nm, 560 nm, and 610 nm. The absorption of 285 nm can be assigned to the π-π* transition of C=C/C=N [30] and the absorption peaks at 560 and 610 nm originated from the n-π* transition of C=N and C=O, demonstrating that there is a large-sized conjugated sp2 domain [31]. From Appendix A, it can be seen that the emission wavelength of the R-CDs in aqueous solution does not change with the excitation wavelength change, and the optimal excitation wavelength is at 560 nm. Under the excitation of 560 nm, two emission peaks caused by the large conjugate sp2 domain can be clearly distinguished at 625 nm and 678 nm (Figure 1b). The quantum yield of the R-CDs in ethanol is calculated to be 22.96% by choosing rhodamine B (QY = 56% in ethanol) as reference. The fluorescence of the prepared R-CDs is tolerant against photobleaching and ionic strength even under the continuous irradiation of excitation light for 180 min and the condition of 1 M NaCl concentration (Appendix A). The particle size distribution and morphology of R-CDs were characterized by DLS and TEM. As can be seen from the DLS data, the particle size of the R-CDs ranged from 2.7 nm to 6.5 nm with an average particle size of 4.0 nm (Figure 2a inset). Spherical R-CDs with good dispersion can be observed clearly from TEM image (Figure 2a). From Figure 1b, we can see that the prepared R-CDs have a wide diffraction peak at about 25°, indicating that they have an amorphous crystal structure [32,33]. The above results show that the developed R-CDs accord with the general characteristics of typical CDs.

The surface-functional groups of the synthesized CDs were characterized by FT-IR. As represented in Figure 2c, the wide peak at 3430 cm^−1^ is considered as the N-H vibration absorption band [34]. The characteristic peaks at 1620 cm^−1^ and 1390 cm^−1^ should be attributed to the stretching vibrations of conjugated structure C=O/C=N and C=C bonds [35]. The absorption peak of C-N bond appeared at 1320 cm^−1^, while a peak at 1127 cm^−1^ is attributed to a C-O stretching vibration. The surface functional groups and elemental composition of R-CDs were further determined by XPS. The full spectrum is shown in Appendix A, the values of the three peaks are 285.11 eV, 400.4 eV, and 531.97 eV, which are assigned to C1s, N1s, and O1s, respectively, and their contents are 80.52%, 15.59%, and 3.91%, respectively. The high-resolution spectrum of C1s (Figure 2d) shows three fitted peaks at 288.1 eV, 285.5 eV and 284.3 eV corresponding to the C=O, C-O/C-N and C=C/C-C bond, respectively. Then, in the N1s high-resolution spectrum of CDs (Figure 2e), three peaks appear at 400.2 eV, 399.8 eV, and 398.9 eV, which belong to the pyrrole N, amine N, and pyridine N. The high-resolution O1s spectrum (Figure 2f) exhibits two different types of O:C=O (532.8 eV) and C-O (531.7 eV). Appendix A shows the values (%) of peak fitted. In summary, the FT-IR and XPS results of R-CDs are consistent. The existence of polar functional groups in R-CDs enables them to disperse well in aqueous solution, which also can promote their interaction with Cr(VI) [36].

### 3.2. Dual-Mode Detection of Cr(VI)

The as-prepared R-CDs possess perfect optical performance and high-stability and can insure the reasonable sensing application. Inspiringly, the developed R-CDs can be an effective dual-mode sensor for the determination of Cr(VI) based on absorption and fluorescent change. As shown in Figure 3a, the absorption of R-CDs solution varies greatly with a color change from blue to faint yellow after Cr(VI) is involved. To realize simple quantitative analysis, scanometric mode was established based the RGB value of the solution image. The absorption change was described as the ratio value of the sum of G and R to B, and the ratio value changes sharply as shown in Figure 3c. On the other hand, Cr(VI) can also cause the fluorescence intensity decrease in R-CDs, which also can be observed easily from the Figure 3b inset. Moreover, in Appendix A, we can find the signals plateaued after 9 min with increasing reaction time. The effective absorption and fluorescence signal change caused by Cr(VI) lays the foundation of the dual-mode quantitative analysis of Cr(VI) (Figure 3c).

The response mechanism of R-CDs to Cr(VI) also was speculated based on a series of experiments, which should be mainly attributed to the unique oxidability of Cr(VI). Firstly, the absorption and fluorescent signals change of R-CDs induced by Cr(VI) must be in acidic conditions (pH = 1~4), and Cr(VI) has a strong oxidizing property at this time. As shown in Appendix A, the involvement of ascorbic acid can significantly hinder the interaction of R-CDs and Cr(VI), and the color, absorption, and fluorescence of R-CDs solution will remain basically unchanged in this case. However, 8-hydroxyquinoline, a strong chelating agent for Cr(VI), has no effect on the dual-mode sensing system after involvement (Appendix A) [7]. Moreover, the average fluorescence lifetime of R-CDs in the absence and presence of Cr(VI) was 2.41 ns and 0.6 ns, respectively, which indicates the occurrence of dynamic quenching (Appendix A). Based on the above results, the dual-mode detection process for Cr(VI) can be mainly explained by the oxidation effect of Cr(VI) to R-CDs.

### 3.3. Analytical Performance

The analytical performance of the proposed dual-mode analysis system was researched carefully. From Figure 4a, the R-CDs solution color changes from blue to faint yellow gradually when increasing the concentration of Cr(VI) that can be distinguished by naked eyes. In this system, 96-well plates were selected as a substrate to support the color reaction and establish the scanometric analysis mode of Cr(VI). Then, the optical photograph was taken in a well-lit area with no obvious shadows by a Smartphone (Huawei HONOR 20). The RGB values of all images taken by a Smartphone were read through the mobile application of Colormeter, and the values of (R + G)/B were calculated to characterize the absorption signal response of R-CDs to different concentrations of Cr(VI). As shown in Figure 4b, the ratio values from the image and the concentrations of Cr(VI) show a good linear relationship in the range of 0.3–50 μM with a correlation coefficient of 0.9971. The limit of detection (LOD) can reach 0.08 μM, which is much lower than the limit value of Cr(VI) in drinking water (~2 μM) [8]. Furthermore, the fluorescent method exhibits higher sensitivity compared to the colorimetric system. From Figure 4c, we can see that the fluorescence intensities of R-CDs present a significant decrease with the increase in Cr(VI) concentrations. The linear range is at 0.03–3 μM between the quenching rate and the concentration of Cr(VI) (r = 0.9965) with a much lower LOD at 9.1 nM, which can ensure the accurate detection of trace Cr(VI) (Figure 4d). More importantly, as shown in Appendix A, the sensitivities of developed methods were comparable and even higher than previous reported CDs-based analysis systems of Cr(VI). Therefore, the scanometric analysis mode can be applied for the quantitative detection of Cr(VI) in environmental water samples, and the fluorescent mode can realize the visual fluorescence imaging of Cr(VI) in vivo.

### 3.4. Selectivity

The specificity of R-CDs to Cr(VI) was checked systematically to facilitate the practical application of established analysis methods. As shown in Figure 5, common metal ions cannot give rise to obvious color and fluorescence change, including Cu^2+^, Cd^2+^, Zn^2+^, Mg^2+^, Fe^3+^, Fe^2+^, Ni^2+^, Hg^2+^, K+, Cr^3+^, Co^2+^, Ca^2+^, Ag^+^, and Al^3+^. Only Cr(VI) can cause the color change from blue to faint yellow of R-CDs solution and red fluorescence quenching, as shown in the Figure 5 inset. Furthermore, some common anions also cannot bring irresistible interference to Cr(VI) detection (Appendix A). All above results indicate that the proposed scanometric and fluorescent method all possesses high-selectivity to meet the requirement of real samples analysis.

### 3.5. Detection of Cr(VI) in Real Water Samples

The application of Cr(VI) detection in environmental water samples was carried out by using tap water, spring water, and lake water samples, which were obtained from the Qilu University of Technology (Shandong Academy of Sciences) laboratory (Jinan, China), Pearl Spring, and Daming Lake (Jinan, China), respectively. Rainwater was collected using a wide-mouth container during a rainstorm. All water samples were spiked with different concentrations Cr(VI) and filtered by using a 0.45 μm filter membrane. The concentrations of Cr(VI) in the original water samples were all lower than the LOD, and the spiked experiments were performed to evaluate the practicability of the methods. In Table 1, the scanometric analysis mode was first applied for the simple and rapid detection of Cr(VI) with spiked concentrations of 1 μM, 10 μM, and 30 μM. The recoveries were between 96.5% and 107.2% and the RSDs were no more than 3.7%. Furthermore, the fluorescence method with a higher sensitivity was used for the trace determination of Cr(VI). At the spiked concentrations of 0.1 μM, 0.5 μM, and 1.5 μM, the Cr(VI) could be accurately detected with recoveries of 92.6–107.2% and RSDs lower than 5% (Appendix A). Therefore, the established scanometric and fluorescence methods have high practical value for detecting Cr(VI) in environmental samples.

### 3.6. Intracellular Imaging of Cr(VI)

The excellent red fluorescence property and high optical stability of synthesized R-CDs provide a new chance for visualizing intracellular Cr(VI). MTT standard assay was carried out to evaluate the toxicity of R-CDs. From Appendix A, we can see that the survival rate still is over 90% even with the concentration of R-CDs at 800 μg·mL^−1^, which proves that the R-CDs have a good biocompatibility. As shown in Figure 6a, the R-CDs can easily penetrate the cell membrane and enter the cell to emit bright red fluorescence. After adding different concentrations Cr(VI), the fluorescence signal in Hela cells decreased gradually (Figure 6b–d). The result suggests that the proposed R-CDs can be used as a new probe for the imaging of intracellular Cr(VI).

## 4. Conclusions

In summary, we presented an effective solid-phase synthesis strategy for preparing R-CDs with high fluorescence and stability by simply pyrolyzing the solid-phase precursor of o-phenylenediamine and aniline hydrochloride. The product yield of this reaction can reach 42.4% while avoiding solvent consumption. The synthesized R-CDs can emit bright red fluorescence with a quantum yield of 22.96%. Furthermore, the R-CDs were successfully applied for the sensitive and selective determination of Cr(VI) through a scanometric and fluorescent dual-mode analysis system. The scanometric mode realized the rapid quantitative detection of Cr(VI) in environmental water samples with satisfactory recoveries of 92.6–107.2%, while the fluorescence system was valid for the visualization imaging analysis of Cr(VI) in vivo. The key significance of this work is the green and facile synthesis of R-CDs and the establishment of scanometric/fluorescent dual-mode sensing to Cr(VI) based on single R-CDs, which is conducive to promoting the commercial application of CDs-based nanomaterials.

## Data Availability

The dataset generated and analyzed in this study is not publicly available but may be obtained from the corresponding author upon reasonable request.

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
