# Peer review of "Solid-Phase Synthesis of Red Fluorescent Carbon Dots for the Dual-Mode Detection of Hexavalent Chromium and Cell Imaging"

_biosensors, 2022, doi:10.3390/bios12060432_

Round 1

Reviewer 1 Report

The authors present and extensive and well-done study on the synthesis of red-emitting CDs and their use as Cr sensor. The characterization demonstrate the important properties of the CDs. The manuscript could be published after a few minor revisions; however, I believe Biosensors is not the adequate journal for it, as it does not involve any biomolecule (either as transducer or analyte). A more general journal or one focusing on chemical sensing should be choosen (e.g., Sensors or Chemosensors, respectively). apart from that, I have a few comments:

* Abstract and conclusion are too similar. Mainly the abstract should be changed to contain more general data. What is the current problem, that the authors want to solve? Why a new synthesis method is required? What is the disadvantage with current Cr sensors? The more quantitative data may then be mentioned in the conclusion.

* section 3.3: the authors mention a "smartphone". A more precise definition should be given, specially regarding the power and if possible, spectrum of the light source used.

* section 3.4: authors should explain the origin/reason for this high selectivity. The functional groups at the surface of CDs seem to be quite general, why does no other metal cause a color/fluorescence change?

Author Response

The authors present and extensive and well-done study on the synthesis of red-emitting CDs and their use as Cr sensor. The characterization demonstrate the important properties of the CDs. The manuscript could be published after a few minor revisions; however, I believe Biosensors is not the adequate journal for it, as it does not involve any biomolecule (either as transducer or analyte). A more general journal or one focusing on chemical sensing should be choosen (e.g., Sensors or Chemosensors, respectively). apart from that, I have a few comments:

RESPONSE: Thanks a lot for your review and it’s our fault that some information is not clear enough in previous version. On the journal official website, the aim of Biosensors is providing an advanced forum for studies related to the science and technology of biosensors and biosensing. In our research, proposed R-CDs was successfully applied for visualizing intracellular Cr(VI) in Hela cells, so we think that our research is suitable for published in Biosensors. Besides, we have revised the manuscript carefully according to the comments. We hope our revision can meet the criteria of publishing, and we are pleased to revise it further if needed.

  1. Abstract and conclusion are too similar. Mainly the abstract should be changed to contain more general data. What is the current problem, that the authors want to solve? Why a new synthesis method is required? What is the disadvantage with current Cr sensors? The more quantitative data may then be mentioned in the conclusion.

RESPONSE: Thanks a lot for your review and we have revised the section of abstract and conclusion according to the comments.

  1. section 3.3: the authors mention a "smartphone". A more precise definition should be given, specially regarding the power and if possible, spectrum of the light source used.

RESPONSE: Thanks a lot for your careful review and it’s our fault that some information is not clear enough. Smartphone has an independent operating system and running space, the equipment that can be provided by a third-party service provider can be installed by the user themselves, and the Smartphone that can be connected to the wireless network. In this study, a Huawei HONOR 20 Smartphone was used which was developed by Huawei Technologies Co Ltd in China, and the mobile phone's own light source is not used. The colorimetric system of R-CDs was carried out in a 96-well plate to ensure the light consistent, and the optical photograph was taken in a well-lit area with no obvious shadows by proposed Smartphone. To make it clear, sentence was added to manuscript:

“Then the optical photograph was taken in a well-lit area with no obvious shadows by a Smartphone (Huawei HONOR 20).” (Page 7, lines 232-233 of the marked revision)

  1. section 3.4: authors should explain the origin/reason for this high selectivity. The functional groups at the surface of CDs seem to be quite general, why does no other metal cause a color/fluorescence change?

RESPONSE: Thanks a lot for your careful review and it’s our fault that some information is not clear enough. We explain the high selectivity of Cr(VI) in section 3.2. The response mechanism of R-CDs to Cr(VI) also was speculated based on series of experiments, which should be mainly attributed to the unique oxidizability of Cr(VI). Firstly, the absorption and fluorescent signals change of R-CDs induced by Cr(VI) must be in acidic conditions (pH=1~4), and Cr(VI) has strong oxidizing property at this time. As show in Figure S5a-c, the involvement of ascorbic acid can significantly hinder the interaction of R-CDs and Cr(VI), and the color, absorption and fluorescence of R-CDs solution basically unchanged in this case. However, 8-hydroxyquinoline, a strong chelating agent for Cr(VI), has no effect on the dual-mode sensing system after involved (Figure S5d-f) [J. Hazard. Mater. 2021, 408, 124898]. Besides, the average fluorescence lifetime of R-CDs in the absence and presence of Cr(VI) was 2.41 ns and 0.6 ns, respectively, which indicates the occurrence of dynamic quenching (Figure S6). Based on the above results, the dual-mode detection process for Cr(VI) can be mainly explained by the oxidation effect of Cr(VI) to R-CDs.

Reviewer 2 Report

The manuscript by Jinshuang Hu describes preparation of Red Fluorescent Carbon Dots for the Dual-mode Detection of Cr(IV). The work is well carried out and presented, however few mnor changes should be addressed before it can be accepted for publication

1) Error bars are missing in some figures

2) In figure 2b, the x axis mislabeled

3) Light conditions required for image acquisition are neither optimized nor specified

4) Linear range for absorbance detection was 0.3-50 μM and for fluorescence detection 0.03-3 μM, therefore the “dual-modesensing system” claimed in the manuscript has the linear range only in the overlapping region (0.03-3 μM). Authors should correct the text accordingly.

5) It is not clear from Materials and Methods section if excessive Cr(IV) was removed from the cell culture after 10min incubation

Author Response

The manuscript by Jinshuang Hu describes preparation of Red Fluorescent Carbon Dots for the Dual-mode Detection of Cr(IV). The work is well carried out and presented, however few mnor changes should be addressed before it can be accepted for publication.

RESPONSE: Thanks a lot for your review! All of these comments have contributed a lot to improve the quality of our manuscript. We have revised the manuscript carefully according to your comments. We hope our revision can meet the criteria of publishing, and we are pleased to revise it further if needed.

  1. Error bars are missing in some figures.

RESPONSE: Thanks a lot for your careful review! Figure S2, S4 and S5 which missed error bars have been added.

  1. In figure 2b, the x axis mislabeled.

RESPONSE: Thanks a lot for your careful review! Figure 2b has been corrected.

  1. Light conditions required for image acquisition are neither optimized nor specified.

RESPONSE: Thanks a lot for your valuable review and it’s our fault that some information is not clear enough. Firstly, the colorimetric system of R-CDs was carried out in a 96-well plate to ensure the light consistent. Then the optical photograph was taken in a well-lit area with no obvious shadows by a Smartphone, and the RGB values of the image were obtained through the mobile application of Colormeter. To make it clear, some sentences were added to manuscript:

Then the optical photograph was taken in a well-lit area with no obvious shadows by a Smartphone (Huawei HONOR 20).” (Page 7, lines 232-233 of the marked revision)

  1. Linear range for absorbance detection was 0.3-50 μM and for fluorescence detection 0.03-3 μM, therefore the “dual-modesensing system” claimed in the manuscript has the linear range only in the overlapping region (0.03-3 μM). Authors should correct the text accordingly.

RESPONSE: Thanks a lot for your careful review. In this research, dual-mode sensing system provides two independent analytical methods, including scanometric detection and fluorescence detection method of Cr(VI). The standard curves of scanometric method (0.3-50 μM) and fluorescence method (0.03-3 μM) were carried out individually. Therefore, the linear range also should be described respectively.

  1. It is not clear from Materials and Methods section if excessive Cr(IV) was removed from the cell culture after 10min incubation.

RESPONSE: Thanks a lot for your careful review and it’s our fault that some information is not clear enough. After different concentrations Cr(VI) (0, 5, 15, 30 μM) were added and incubated for 10 min, we use fresh buffer solution to wash twice. To make it clear, some sentences were rephrased/added to manuscript:

“Before imaging, the cells were washed twice with fresh buffer solution to remove excessive Cr(VI). The images were captured again using a fluorescence microscope under green laser light excitation.” (Page 4, lines 143-145 of the marked revision)

Reviewer 3 Report

Dear Authors, 

This manuscript presents Solid-phase Synthesis of Red Fluorescent Carbon Dots for the Dual-mode Detection of Hexavalent Chromium and Cell Imaging. This work results are well presented.

In Fig. 2 the XPS fitting results should be provided in a form of Table in the Supporting Information, given as % values for each of the peak fitted.

Author Response

Dear Authors, This manuscript presents Solid-phase Synthesis of Red Fluorescent Carbon Dots for the Dual-mode Detection of Hexavalent Chromium and Cell Imaging. This work results are well presented.

RESPONSE: Thanks a lot for your review! We have revised the manuscript carefully according to your comment and we are pleased to revise it further if needed.

  1. In Fig. 2 the XPS fitting results should be provided in a form of Table in the Supporting Information, given as % values for each of the peak fitted.

RESPONSE: Thanks a lot for your careful review! The XPS fitting results have been added in a form of Table S1 in the Supporting Information. To make it clear, sentence was added to manuscript:

“Table S1 shows the values (%) of peak fitted.” (Page 5, lines 184 of the marked revision)

Table S1. Values (%) of each peak fitted

Element

Peak Fitted

Value (%)

C

C=O

34.2

C-O/C-N

33.5

C-C/C=C

32.3

N

Pyrrole N

5.3

Amine N

53.4

Pyridine N

41.3

O

C=O

34.2

C-O

65.8